# The Intra-Hospital Medical Dispute Burden and Capacities: A Nationwide Survey in Taiwan

**DOI:** 10.3390/healthcare11152121

**Published:** 2023-07-25

**Authors:** Wen-Chun Chia, Li-Sheng Chen, Sen-Te Wang

**Affiliations:** 1Department of Family Medicine, Taipei Medical University Hospital, Taipei 110, Taiwan; 213048@h.tmu.edu.tw; 2School of Oral Hygiene, College of Oral Medicine, Taipei Medical University, Taipei 110, Taiwan; samchen@tmu.edu.tw; 3Department of Family Medicine, School of Medicine, College of Medicine, Taipei Medical University, Taipei 110, Taiwan

**Keywords:** medical disputes, mediation, alternative dispute resolutions, communication, nationwide survey

## Abstract

(1) Background: Medical disputes have long been resolved via lawsuits. Alternative dispute resolutions have been promoted for their benefits and win–win results. This study aims to investigate Taiwanese hospital medical dispute capacities and burdens. (2) Methods: This study used 2015 nationwide questionnaire data. The number and value of medical disputes that occurred in 2014 was examined to evaluate hospitals’ capabilities. Poisson regressions were used to determine the impact of coping abilities on the incidence of disputes and the associated compensation. (3) Results: The response rate of the questionnaire was 90%. Hospital features associated with higher medical disputes incidence included those of a scale ≤ 100 or 200–499 and having a dispute–inform process of over 4 h. In contrast, hospitals whose compensation fund was solely based on medical liability insurance reported less medical dispute incidence. The features associated with higher compensation were lack of continuing training and having a dispute–inform process over 4 h. In contrast, hospitals with standard operating procedures for in-hospital mediation and solicitude paid lower compensation. (4) Conclusions: Hospitals with quicker response times experienced fewer medical disputes and paid lower compensation. Dispute coping skills, other than reaction time, were more visible in compensation bargaining, but were not significantly correlated with incidence.

## 1. Introduction

A medical dispute occurs when a patient seeks an apology or reimbursement [1,2], which may arise from misunderstandings due to poor communication, or medical malpractice resulting from negligence or iatrogenic harm. Resolving these disputes can involve negotiation via full communication, third-party mediation, or litigation [3]. In Taiwan, individuals can resolve medical disputes through private negotiation or through third-party mediation and can simultaneously call for aid from various sources. When the negotiation breaks down, litigation is the last resort due to its lengthy, costly, and frustrating process [4]. For similar reasons, and considering the adverse impact [3,4,5] of antagonism [3,6,7] between patients and the medical parties during lawsuits, experts strongly advocate for system and regulation reforms, such as no-fault liability or compensation, a dual-track system [3,8,9], and alternative dispute resolutions (ADRs). ADRs, mediation, or communication-and-resolution programs [10,11,12] aim to achieve a win–win outcome. Nowadays, worldwide healthcare communities increasingly emphasize this alternative management for patient safety and to prevent medical disputes.

Adopting the concept of a disease prevention model, the strategies for preventing and addressing medical disputes can be classified into three tiers. The primary tier aims to promote an organizational culture that enhances personnel’s knowledge and awareness and endorses patient safety [13,14]. It also involves proactive risk assessment using professional techniques. The secondary tier aims to identify the weaknesses in organizational aspects, procedures, and personnel, and to rectify them promptly via blame-free disclosure, liability mechanisms, and critical incident reviews [15]. The tertiary tier focuses on post-dispute healing, tragedy control, psychological recovery, and case examination for both patients/families and personnel. The timing is classified into two phases: the early stage, which aims to prevent disputes from escalating into lawsuits, and the late stage, which offers support to both patients and personnel following a lawsuit.

Though there have been numerous previous studies evaluating historical medical disputes, many of them have focused on the individual personnel or specific departments [2,16] or have been based on cases that skipped or failed to undergo early-stage mediations [4]. The Institute of Medicine (IOM) report and the Swiss cheese model suggest that most medical malpractice results from inevitable human error in a flawed system, rather than solely from individual negligence [17].

It is valuable for national health authorities to understand the hospital-level factors associated with higher medical dispute occurrence and unfavorable negotiation outcomes. To understand the current situation and the hospital characteristics related to increased medical dispute incidence and compensation, a nationwide questionnaire survey was conducted. In addition, we aimed to gain a deeper understanding of medical dispute prevention, coping, and management. Our hypothesis is that hospitals with well-established policies are better at coping with medical disputes, avoiding high compensation, facilitating the integration of historical dispute experience into intra-institute professional training, and ultimately achieving a lower medical dispute incidence in the long term.

## 2. Materials and Methods

### 2.1. Questionnaire Investigation

The questionnaire was designed by three experts in the fields of medical ethics, social medicine, and public health. The questionnaire was reviewed and the contents independently validated by three expert reviewers. The inclusion of hospitals in the study was based on the requirement that all hospitals in Taiwan participate in hospital accreditation. Patient safety and medical dispute issues are part of the evaluation criteria, and the questionnaire survey covered all hospitals. Psychiatric hospitals were excluded from the analysis because they are specialized institutions with unique characteristics. The focus of our study was on general hospitals, and therefore we made the decision to exclude psychiatric hospitals for the purpose of our analysis.

The questionnaires were distributed to hospitals from August to October in 2015, with a return envelope enclosed. The survey aimed to investigate the burden of hospitals with medical disputes and associated compensation amounts and to understand the features of heavy-burden hospitals and their coping capabilities. The final questionnaire is composed of 30 items: hospital attributes (4 items), dispute coping competence training program (3 items), existing dispute processing and coping capacities (13 items) (e.g., holding standard operating procedure (SOP), with a chairman elected from the top of the intra-hospital administrative hierarchy, committee meeting frequency, and intra-institute informing efficiency), historical dispute attributes (3 items), and the current burden of medical disputes (7 items).

Since numerous headings are used for entities in charge of in-hospital mediation and solicitude (IMS) across hospitals, a definition of entities responsible for medical dispute coping was listed in the questionnaire as: “staff or intra-hospital task forces responsible for promptly communicating with patients and their families or agents to clarify disputes, address concerns, alleviate emotions, coordinate, and provide aid in rehabilitation if needed.” Their primary objective is to prevent disputes from escalating into litigation and achieve smooth settlement agreements. In addition, to maintain uniformity of questionnaire response across hospitals, adjustments were made to the items, followed by deliberation on the possible pathways for transferring and the units responsible for filling it out and sending it back after it was received by the hospital mailroom. These adjustments and deliberations were initially based on the experience gained from official communication and further consulted various internal expert committees of the Ministry of Health and Welfare (MHW).

The investigation required determining the amounts of outpatient and hospitalizations in 2015. To diminish the respondents’ intention to under-report and make their dispute incidence inclined to the national average, the data on service amounts were obtained from the public data of monthly statistics on the claims of National Health Insurance in Taiwan. In addition, the denominators for calculating questionnaire response rates were derived from the annual report on the status and service statistics of medical institutions in 2015.

### 2.2. Statistical Analysis

The study data were summarized as mean (standard deviation, SD) or counts (percentage) unless otherwise specified. Significant factors were included in the Poisson regression model to simultaneously assess the significant hospital-level factors associated with medical dispute incidence and the associated compensation. Among all the hospital characteristics, particular attention was given to those related to the existence and adequacy of IMS groups’ operation mechanisms. Throughout this article, two-tailed tests with a significance level set at 0.05 are used to evaluate statistical associations. The analyses were performed using SAS release 9.2 (SAS Institute Inc., Cary, NC, USA).

## 3. Results

### 3.1. Summary of Nationwide Hospitals and the Questionnaire Survey Response

Of the 494 registered hospitals surveyed in 2015, 43 psychiatric hospitals were excluded from this survey because of their specific nature. The questionnaires were distributed to the remaining 451 hospitals and returned by 404 hospitals (response rate: 90%). At the time of the survey, the 494 hospitals were composed of 16% public institutes, 24% established by private corporation aggregates, 5% affiliated with private institutes (e.g., schools, associations), and 55% privately owned sole proprietorship. The response rates of the four categories of hospitals were 92%, 100%, 25%, and 88%, respectively. When considering hospital levels, 91% response rates were observed for medical centers and regional or district hospitals, while hospitals without accreditation had a low response rate of 15%.

### 3.2. The Burdens Resulted from Medical Disputes, According to Experience from Historical Cases

An overview of historical medical disputes is presented in Table 1: 33.89% was attributed to worse communication or attitude, 19.22% to medical risks, and 7.33% to unexpected causes. Based on the self-reported rate of disputes leading to violence (about 20%), it can be inferred that on average, five disputes may potentially result in one act of violence.

### 3.3. The Medical Dispute Coping Capacities of Hospitals

Surveys on the medical dispute coping capacities were summarized in Table 2. Regarding safety culture cultivation, the majority of hospitals made historical dispute management documents retrievable (78.07%), conducted relevant continuing training programs (80.3%), and established post-dispute reviewing mechanisms (90.48%). The most common length of training courses was 1–2 h (92.45%). However, it is noteworthy that a small proportion of hospitals (3.51%) reviewed the disputes to clarify the causes.

Regarding dispute coping entities, the majority of hospitals (90.84%) held IMS groups. Among hospitals with IMS groups, 96.73% had a chairperson served by a senior executive, 97% held SOP for the group running modes, while 77% held meetings on a case-by-case basis, 22% held regular meetings, and 78% maintained retrievable meeting records. Among hospitals with regular meetings, 52% held them quarterly, and 95% held them at least semiannually. This indicated that about 77% of IMS entities operated in a formalistic manner during peacetime rather than maintaining steady operation.

Regarding compensation and financial support, of the 404 hospitals, 53.13% had medical liability insurance to handle financial compensation for medical disputes, 29.82% had an intra-hospital mutual aid fund, and 25.06% did not have a regular fund for compensation.

In terms of dispute coping performance, 25.82% of hospitals were able to resolve disputes within an average duration of 1 week, 39.84% resolved them within 1 month, 21.15% resolved them within 1–3 months, and only a minority (5.49%) had an average resolution time over 6 months. Regarding holding coping meetings after disputes occurred, most hospitals (91.58%) held them within 1 week on average. More than half of hospitals (64.69%) successfully managed to resolve over 80% of medical disputes without third-party mediation or litigation. However, 12.46% of hospitals seemed to have a lower capacity, as their reported success rate in achieving resolutions was under 40%.

### 3.4. The Medical Dispute Burden of Hospitals

A cross-sectional overview of the disputes that occurred in 2014 was summarized in Table 3. Of the 404 hospitals that responded, 49.26% (199/404) had experienced at least one medical dispute, resulting in a total of 1392 cases. This indicates an average annual incidence rate of three cases per hospital. Among the 199 hospitals, only a small minority (2.01% = 4/199) were unable to resolve the dispute via IMS systems.

Out of the 1392 newly occurred medical disputes in 2014, 81.68% (1137/1392) were handled through IMS systems, with 61.06% (850/1392) successfully resolved. Additionally, 9.84% (137/1392) were resolved via out-of-hospital mediation, and 15.95% (222/1392) resorted to litigation (civil litigation: 4.96%, criminal litigation: 9.41%, criminal activity/supplementary civil action: 1.58%).

Including disputes prior to 2014, 29.6% of hospitals (119/404) were involved in litigation, with an average of about one case per hospital (401/404). Among these hospitals, 53% (62/119) had two or more disputes resulting in litigation. As of August 2015, 45.64% of the litigation cases had occurred in 2014; while the remaining 54.36% (218/401) had lasted for at least 1.7 years.

### 3.5. Hospital Characteristics Related to Medical Dispute Incidence and the Associated Compensation

The Poisson regression analysis of the 1392 newly incident medical disputes in the 404 response hospitals is summarized in Table 4. Hospital features associated with significantly higher medical dispute incidence included: having a scale in for a number of beds ≤ 100 (RR [95% CI]: 4.62 [2.24–9.51] for <50 beds; 5.2 [2.54–10.62] for 50–99 beds) or in 200–499 (1.98 [1.46–2.69]), being a subsidiary of private institutes (3.16 [1.98–5.04]), and taking over 4 h for an intra-hospital dispute–inform process (1.66 [1.32–2.08]). Conversely, hospitals associated with a lower incidence of medical disputes were those that had medical liability insurance as part of their compensation fund (0.64 [0.45–0.93]). Hospitals more likely to face significantly higher compensation were those that lacked continuing training courses on medical dispute coping competence (6.74 [2.82–16.10]) and took more than 4 h to accomplish intra-hospital dispute inform (2 [1.08–3.71]). In contrast, hospitals without a post-dispute reviewing mechanism (0.25 [0.07–0.87]) and hospitals holding IMS groups with an SOP (0.08 [0.02–0.27]) observed lower compensation amounts.

## 4. Discussion

This study utilized nationwide survey data from MHW in 2015 to investigate the medical dispute burden of hospitals and features associated with higher incidence and compensation. In summation, this study investigated the current infrastructure for resolving medical disputes within hospitals and through legal means. The findings suggested that (i) the hospitals with more active and efficient responses to medical disputes experienced lower incidence rates and lower compensation; (ii) the capacity of hospital dispute coping, with the exception of the reaction time, was found to be more prominent in post-dispute compensation bargaining, while it did not show a significant correlation with dispute incidence; (iii) IMS entities play a significant role in post-dispute resolutions, especially in compensation negotiations, but more progress is needed in integrating dispute-coping experience feedback into pre-dispute prevention efforts.

Our study differs from previous analyses in three key aspects. First, numerous studies analyzed verdicts from the national judicial system database and focused on case characteristics at the late stage [18,19,20]. Observations based on litigation verdicts highlighted cases in surgery and obstetrics, especially those resulting in patient death. In contrast, our study investigates an earlier stage of medical disputes [4,10]. About one-third of medical disputes were attributed to poor communication and service attitude, emphasizing the need for enhanced communication competence and training in response to the growth of consumerism. Second, most studies have focused on the features of individual medical personnel [16] or departments with a high risk of medical dispute [21,22]. In contrast, our study diverges by examining the organizational safety climate and culture [23,24,25], e.g., the intention to disclose medical incident [26]. Third, numerous tools have been developed to assess the safety climate [27,28,29], but still focus on individual personnel. In contrast, our study provides a comprehensive overview of the intra-hospital entities and operational mechanisms that contribute to pre-dispute prevention, post-dispute coping, and feedback from historical disputes.

Our study findings revealed that smaller and medium-sized hospitals had a higher incidence of medical disputes, contrary to the findings of a verdict study [4] that suggested larger hospitals had a higher incidence of claims, although clinics had the highest percentage of paid claims. Additionally, a litigation study on Anesthesia and Intensive Care Units [30] demonstrated that smaller hospitals often face a greater number of claims. Conversely, a mediation study [12] highlighted that larger hospitals specializing in treating patients with serious conditions were more likely to be involved in complex medical disputes with higher compensation. However, our study did not find any association between hospital volume and compensation. In terms of handling time, our study revealed that the dispute–inform process taking over four hours was associated with higher incidence of medical dispute and compensation. This aligns with the results of a mediation study [12] that showed a 0.2–0.3% increase in compensation for each additional day of duration. Furthermore, our study findings indicated that hospitals with medical liability insurance as part of their compensation fund had a lower incidence of medical disputes. This contrasts with the findings of Luo et al. [31], who found that medical liability insurance coverage was not significantly associated with medical disputes. However, the lack of current evidence regarding other significant findings necessitates further study in the future.

ADR practices include mediation, facilitated settlements, and conflict management, offering alternative dispute resolution methods to avoid litigation [32]. According to Wang et al., mediation effectively resolves disputes, with an 89% success rate and an 87-day average duration [12]. In our study, we focused on IMS, which has been actively promoted by MHW of Taiwan from 2013 as a more recommended ADR approach. It involves mediations through communication, apology, and commitment to learning from past incidents to enhance patient safety in the future [33,34,35,36]. Similar IMS entities have been widely implemented in countries other than Taiwan. ADR deserves the attention of policy makers due to its potential economic benefits and time savings compared to traditional lawsuits [37,38]. What makes ADR attractive is that it can be implemented by medical societies without legal reforms [10].

Two recommendations are proposed for future research. Firstly, there is a need to unify the official mediation database and enhance its profiles with comprehensive information from the initiation of a case to the conclusion of mediation [33,34]. This will facilitate the generation of knowledge crucial for the development and maintenance of effective out-of-court resolutions, particularly in the context of ADR. Secondly, it is essential to closely monitor the operations of IMS entities and promote their role in establishing a feedback loop between dispute coping experiences and ongoing intra-hospital training programs.

Some limitations of our study are discussed below. Firstly, psychiatric hospitals, which operate differently from other hospitals, were completely excluded from this study. Private hospitals and those without accreditation exhibited low response rates. Secondly, due to regulations regarding data confidentiality, it was not possible to evaluate whether the hospital attributes were associated with the probability of achieving settlements (i.e., successful mediations) in this study. Conducting such an evaluation would require aligning individual case profiles with hospital responses in questionnaires or obtaining case profiles via the questionnaire survey, which would violate the principle of not tracing individual dispute cases under any circumstances.

## 5. Conclusions

In summary, this study investigated the existing infrastructure of intra-hospital and out-of-court resolutions for medical disputes. The findings suggested that hospitals that reacted more promptly to medical disputes experienced lower dispute incidence and lower compensation. Furthermore, the capacity of hospitals to cope with disputes, apart from the reaction time, was more evident in post-dispute compensation bargaining but did not show a significant correlation with dispute incidence.

## Figures and Tables

**Table 1 healthcare-11-02121-t001:** The characteristics of response hospitals in the questionnaire survey (*n* = 404).

Characteristics	*n*	%
Hospital attributes		
Regulatory authority/funding source category		
Public institutes	67	16.83
Private medical corporation aggregate	107	26.88
Private institutes (e.g., school, associations)	6	1.51
Private sole proprietorship hospital/sole corporation	218	54.77
No response	6	
Hospital levels		
Medical center	20	4.96
Regional hospital	71	17.62
District hospital	305	75.68
Hospitals without accreditation	7	1.74
Unknown	1	
Hospital scale (beds, including acute beds, psychiatric beds, chronic beds)		
49 or less	177	43.81
50–99	63	15.59
100–199	25	6.19
200–499	83	20.54
500 or more	56	13.86
The profile of historical medical disputes		
The disputes category (multiple choice)		
Medical care	257	28.56
Poor communication	179	19.89
Medical risks	173	19.22
Service attitude	126	14
Drug safety	86	9.56
Unexpected public security	66	7.33
Others	13	1.44
Disputes-induced violence		
None	312	80
>0, ≤10%	67	17.18
>10, <30%	6	1.54
>30%	5	1.28
No response	14	-

**Table 2 healthcare-11-02121-t002:** The characteristics regarding dispute coping capacity of hospitals (*n* = 404).

Characteristics	*n*	%
Medical dispute coping competence training and management		
The dispute handling records are retrievable		
Yes	299	78.07
No	84	21.93
No response	21	-
A post-dispute reviewing mechanism has been implemented		
No response	26	-
No	36	9.52
Yes	342	90.48
The reviewing focus is on dispute causes	*12*	*3.51*
The reviewing focus is on dispute coping strategies	*240*	*70.18*
The reviewing process is documented and retrievable	*270*	*78.95*
Are there ongoing training programs with respect to dispute prevention and coping?		
No response	8	
No such courses are held	78	19.7
Yes, such courses are held	318	80.3
The average course length in hours		
About 1 h	219	68.87
About 2 h	75	23.58
About 3 h	9	2.83
About 4 h	11	3.46
More than 4 h	4	1.26
Dispute processing and coping mechanism		
Compensation fund system		
Medical liability insurance	180	45.11
Intra-hospital mutual aid fund	87	21.8
Both of the above	32	8.02
None	100	25.06
No response	5	
Are there existing IMS groups for medical dispute coping?		
No	37	9.16
Yes	367	90.84
Do the IMS groups maintain their SOPs?		
Yes	356	97
No	10	2.72
The IMS groups’ running modes (multiple choice)		
Regularly hold meetings	85	21.68
Annually	4	4.71
Semiannually	25	29.41
Quarterly	44	51.76
Monthly, weekly, or other frequency	12	14.12
Hold meetings by case	301	76.79
Others	6	1.53
The IMS group members consist of:		
Serves as the chairperson	355	96.73
The IMS group meeting record is retrievable		
Yes	288	78.47
No	72	19.62
No response	7	1.91
The average processing duration whenever a dispute arises		
From the time the dispute first arises to the time at which hospital authorities are informed		
No more than 4 h	269	70.79
4 to 8 h	50	13.16
8 to 24 h	53	13.95
More than 24 h	8	2.11
No response	24	-
Time from dispute arising to a coping meeting being held		
≤3 days	234	61.58
≤1 week	114	30
≤1 month	28	7.37
>1, ≤3 months	1	0.26
>3, ≤6 months	3	0.79
No response	24	-
Time from dispute arising to a resolution being achieved		
≤3 days	28	7.69
≤1 week	66	18.13
≤1 month	145	39.84
>1, ≤3 months	77	21.15
>3, ≤6 months	28	7.69
>6 months	20	5.49
No response	40	-
Achieving resolution after IMS groups intervention		
≤10%	20	5.93
>10, ≤40%	22	6.53
>40, ≤60%	29	8.61
>60, ≤80%	48	14.24
>80%	218	64.69
No response	67	

Note: IMS = in-hospital mediation and solicitude. SOP = standard operating procedures.

**Table 3 healthcare-11-02121-t003:** Summary for the burden of medical disputes.

Characteristics	*n*	%
An overview of cases that occurred in 2014		
New medical dispute cases in responding hospitals		
At least 1 case	199	49.26 ^[c]^
Total cases (average over responding hospitals)	1392	(3.4) ^[c]^
New cases entered the IMS system of the hospital		
At least 1 case (% of hospitals with at least 1 dispute case)	195	97.99 ^[c]^
Total cases (% of total cases of response hospitals)	1137	81.68 ^[a]^
Case consequences		
Resolved under IMS system (%)	850	61.06 ^[a]^
Compensation required (%)	332	39.06 ^[b]^
Resolved via mediation by external institutes/organizations (%)	137	9.84 ^[a]^
Compensation required (%)	70	51.09 ^[b]^
Resorting to civil litigation (%)	69	4.96 ^[a]^
Compensation required (%)	11	15.94 ^[b]^
Resorting to criminal litigation (%)	131	9.41 ^[a]^
Compensation required (%)	4	3.05 ^[b]^
Resorting to criminal activity/supplementary civil action (%)	22	1.58 ^[a]^
Litigation burden from historical disputes		
Litigation cases that remained unresolved by the end of August 2015 ^†^		
No response	2	-
None	283	70.4 ^[c]^
At least 1 case	119	29.6 ^[c]^
Only 1 case	55	47.01
2–10 cases	54	46.15
More than 10 cases	8	6.84
Total cases (average over response hospitals)	401	(1) ^[c]^

NOTE: SD = standard deviation. †: Dispute cases that occurred before 2014 were included. **a**: The denominator of the percentage is the total number of medical dispute cases in response hospitals. **b**: The denominator of the percentage is the total number of cases that satisfied the category. **c**: The statistics listed in this row are summaries over the 404 response hospitals.

**Table 4 healthcare-11-02121-t004:** The Poisson regression analysis results for the medical disputes incidence and compensates.

	Medical Disputes Incidence		The Associated Compensates	
	Multivariant		Multivariant	
	aRR	95% CI	aRR	95% CI
Regulatory authority/funding source category				
Public institutes	0.85	0.58–1.25	0.62	0.23–1.67
Private medical corporation aggregate	0.85	0.59–1.21	0.62	0.23–1.65
Private institutes (e.g., school, associations)	3.16	1.98–5.04	0.14	0.01–1.48
Private sole proprietorship hospital/sole corporation	1.00		1.00	
Hospital levels				
Medical center	1.44	0.03–65.70	1.77	0.01–424
Regional hospital	1.41	0.03–63.83	1.13	0.00–257
District hospital	0.65	0.01–28.56	0.22	0.00–46
Hospitals without accreditation	1.00		1.00	
Hospital scale (beds)				
49 or less	4.62	2.24–9.51	0.69	0.14–3.48
50–99	5.20	2.54–10.62	1.21	0.22–6.76
100–199	2.38	0.93–6.06	2.06	0.37–11.46
200–499	1.98	1.46–2.69	1.29	0.56–2.98
500 or more	1.00		1.00	
Compensation fund system				
Medical liability insurance	0.64	0.45–0.93	0.96	0.38–2.42
Mutual aid fund	1.06	0.77–1.45	0.94	0.42–2.10
Both	0.95	0.63–1.45	1.44	0.49–4.25
No	1.00		1.00	
IMS mechanism for coping medical disputes				
Yes (with SOP)	0.69	0.23–2.11	0.08	0.02–0.27
Yes (without SOP)	0.48	0.07–3.20	0.14	0.01–1.55
No	1.00		1.00	
Duration from occurrence of the dispute to the hospital being informed				
More than 4 h	1.66	1.32–2.08	2.00	1.08–3.71
Less than 4 h	1.00		1.00	
Is there a post-dispute reviewing mechanism in place?				
No	0.77	0.37–1.60	0.25	0.07–0.87
Yes	1.00		1.00	
Are there ongoing training courses?				
No	0.84	0.51–1.40	6.74	2.82–16.10
Yes	1.00		1.00	

## Data Availability

The datasets used and/or analyzed during the current study are available from the corresponding author on reasonable request.

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
