# Peer review of "The Intra-Hospital Medical Dispute Burden and Capacities: A Nationwide Survey in Taiwan"

_healthcare, 2023, doi:10.3390/healthcare11152121_

Round 1

Reviewer 1 Report

Medical disputes are an important issue that is occurring with increasing frequency in healthcare. The analysis made shows that good dispute management can bring many benefits to the hospital but also to patients. I believe that the work is very important in economic terms. It is prepared well in terms of statistics and cleanly presents the results.

The methodology presented in the paper is well done. It contains all the elements that a well-prepared methodology must contain. The results presented in the work are relevant to the current state of health care and can translate into the current system if appropriate conclusions are drawn from them.

The number of hospitals surveyed is impressive. The authors put in a lot of time and effort to compile such a large number of studies.

I have no reservations about the content of the article.

Author Response

Dear Reviewer,

Thank you for your feedback and suggestions regarding our manuscript. We appreciate your careful review and are glad to hear that the issues raised have been addressed satisfactorily. We would like to express our gratitude for your valuable input, which has undoubtedly contributed to the improvement of our work.

We are pleased to hear that you found our methodology well-designed and comprehensive. We strived to ensure that all essential elements of a robust methodology were incorporated into our study.

Once again, we sincerely appreciate your thoughtful review and positive feedback. We are confident that our manuscript contributes valuable knowledge to the field, and we are grateful for the opportunity to share our findings with the scientific community.

Sincerely,

Wen-Chun Chia

Reviewer 2 Report

This article is a report of a survey intended to identify the organizational culture and its impact (and costs) on resolving medical disputes. The findings may be important and new for Taiwan, but I doubt that an international audience would find the study surprising or helpful. Better IMS cultures result in fewer costs for the institution. What would be interesting is to explore more deeply that ADR options that would work for Taiwan, but ADR receives little discussion.   

I do not understand the first sentence: "A medical dispute is a patient proposal in return for the argument which could be requests in any form, usually an apology or reimbursement." My guess is that the problem lies with poor word choice somewhere in the following: "return for the argument which could be requests in any form." The rest of the first paragraph also needs grammar editing, but the meaning is clearer. The same can be said for the rest of section one. 

Section two had some minor grammar errors. Readability could be improved by shortening extra long sentences. 

Section three had many word choice and word form errors, which made reading difficult. 

Grammar issues in section four obscured some of the authors' meaning. 

Author Response

Dear Reviewer,

Thank you for your feedback on our article. We appreciate your comments and suggestions.

Q1: What would be interesting is to explore more deeply that ADR options that would work for Taiwan, but ADR receives little discussion.   

Ans: In the discussion section, we have expanded upon the topic of ADR and included references to relevant studies. “ADR practices include mediation, facilitated settlements, and conflict management, offering alternative dispute resolution methods to avoid litigation[32]. According to Wang et al., mediation effectively resolves disputes, with 89% success rate and 87-day average duration[12].”

Q2: I do not understand the first sentence: "A medical dispute is a patient proposal in return for the argument which could be requests in any form, usually an apology or reimbursement." My guess is that the problem lies with poor word choice somewhere in the following: "return for the argument which could be requests in any form." The rest of the first paragraph also needs grammar editing, but the meaning is clearer. The same can be said for the rest of section one.

Section two had some minor grammar errors. Readability could be improved by shortening extra long sentences.

Section three had many word choice and word form errors, which made reading difficult.

Grammar issues in section four obscured some of the authors' meaning.

Ans: Regarding your comment on the first sentence, we apologize for the confusion caused by poor word choice. The sentence should have been clearer and more concise. We have revised it to improve its clarity.

We have also taken your suggestions into account regarding grammar editing and readability in sections one, two, three, and four. We have made necessary revisions to ensure the overall coherence and understandability of the content.

Once again, we would like to express our gratitude for your valuable input. Your feedback has greatly contributed to the improvement of our article.

Sincerely,

Wen-Chun Chia

Reviewer 3 Report

The authors must detail the method used, including the questionnaire and its validity, the inclusion criteria, and the reason for excluding psychiatric hospitals. 

The discussion section must be improved. To better understand the findings, utilize literature on organizational behavior and culture. Furthermore, the results require a more detailed discussion, for example, the conclusion that small and large hospitals are more prone to disputes than medium-sized hospitals. While the recommendation of promptly addressing disputes and providing training is generic, the evidence must support it. Similarly, other results necessitate more elaboration and should be compared to the current state of the art. The authors must provide more context and detail on significant findings.

It is important for authors to use recent and relevant references in their work, as many of the current references are 5 to 10 years old or even older. When drawing conclusions, it is advised to be cautious and take this into consideration. 

No major issue. 

Author Response

Dear Reviewer,

We would like to express our sincere gratitude for your valuable feedback on our article. We have carefully considered your comments and have made significant improvements based on your suggestions.

Q1: The authors must detail the method used, including the questionnaire and its validity, the inclusion criteria, and the reason for excluding psychiatric hospitals. 

Ans: Regarding the methodology, we have provided detailed information on the questionnaire used, its validity, the inclusion criteria, and the rationale for excluding psychiatric hospitals. We believe these additions will enhance the transparency and rigor of our study. “All questions in the questionnaire were designed by three experts in the fields of medical ethics, social medicine, and public health. The questionnaire was reviewed and validated the contents by three of the expert reviewers. The experts conducted their re-view independently. The inclusion of hospitals in the study was based on the requirement for all hospitals in Taiwan to participate in hospital accreditation. Patient safety and medical dispute issues are part of the evaluation criteria, and the questionnaire survey covered all hospitals. Psychiatric hospitals were excluded from the analysis because they are specialized institutions with unique characteristics. The focus of our study was on general hospitals, and therefore we made the decision to exclude psychiatric hospitals for the purpose of our analysis.”

Q2: The discussion section must be improved. To better understand the findings, utilize literature on organizational behavior and culture. Furthermore, the results require a more detailed discussion, for example, the conclusion that small and large hospitals are more prone to disputes than medium-sized hospitals. While the recommendation of promptly addressing disputes and providing training is generic, the evidence must support it. Similarly, other results necessitate more elaboration and should be compared to the current state of the art. The authors must provide more context and detail on significant findings.

Ans: In response to your comments on the discussion section, we have incorporated relevant literature on organizational behavior and culture to provide a better understanding of our findings. We have also expanded the discussion to provide more detailed insights into the specific conclusions, such as the relationship between hospital size and the incidence of disputes. Additionally, we have provided more context and detail on significant findings, comparing them to the current state of the art. “Our study findings revealed that smaller and median-sized hospitals had a higher incidence of medical disputes contrary to the findings of a verdict study [4], that suggested larger hospitals have a higher incidence of claims, although clinics had the highest percentage of paid claims. Additionally, a litigation study in Anesthesia and Intensive Care Units [30] demonstrated that smaller hospitals often faced a greater number of claims. Conversely, a mediation study [12] highlighted those larger hospitals, specializing in treating patients with serious conditions, were more likely to be involved in complex medical disputes with higher compensation. However, our study did not find any association between hospital volume and compensation. In terms of handling time, our study revealed that dispute-inform process taking over four hours was associated with higher incidence of medical dispute and compensation. This aligns with the results of a mediation study [12] that showed an increase in compensation by 0.2 to 0.3% for each additional day of duration. Furthermore, our study findings indicated that hospitals with medical liability insurance as part of their compensation fund had a lower incidence of medical disputes. This contrasts with the findings of Luo et al. [31], who found that medical liability insurance coverage was not significantly associated with medical disputes. However, the lack of current evidence regarding other significant findings necessitates further study in the future.”

Q3: It is important for authors to use recent and relevant references in their work, as many of the current references are 5 to 10 years old or even older. When drawing conclusions, it is advised to be cautious and take this into consideration. 

Ans: We acknowledge your concern about the references used in our work. We have revised our reference list to include more recent and relevant sources.

Once again, we would like to express our appreciation for your valuable input. Your feedback has significantly improved the quality and clarity of our article.

Sincerely,

Wen-Chun Chia

Round 2

Reviewer 2 Report

The English editing greatly improved the paper, and the discussion section revisions are helpful. 

I would suggest (but not require) one more close check for grammar and readability to polish it. 

Author Response

Dear reviewer,

Thank you for your valuable feedback. We conducted one more thorough check for grammar and readability to further refine the paper. Your comment is appreciated.

Wen-Chun, Chia.

Reviewer 3 Report

The authors have significantly improved the manuscript and effectively addressed previous feedback.

Author Response

Dear reviewer,

Thank you for your valuable feedback. We have carefully addressed all previous feedback and made significant improvements to the manuscript. Your comment has been instrumental in enhancing the quality of our work.

Wen-Chun, Chia